

# Undernutrition and feeding difficulties among children with disabilities in Zambia: insights for inclusive nutrition strategies

Zeina Makhoul[1], Julie M. Long[2], Mulemba Ndonji[3], Carolyn Moore[1], Watson Shungu Mwandileya[3], Edgar Lunda[3], Kate Miller[1], Bradley S. Miller[4] and Douglas Taren[1,2]

[1] SPOON, Portland, OR, United States of America
[2] Section of Nutrition, Department of Pediatrics, University of Colorado School of Medicine, Aurora, CO, United States of America
[3] Access to Health Zambia, Lusaka, Zambia
[4] Division of Endocrinology, Department of Pediatrics, University of Minnesota Medical School, Minneapolis, MN, United States of America

Corresponding author
Zeina Makhoul,
zeina@spoonfoundation.org

## ABSTRACT

**Background.** Understanding the burden of undernutrition in children with disabilities is crucial for designing inclusive nutrition programs and policies. This study aimed to estimate the prevalence of undernutrition and risk for feeding difficulties and describe common feeding practices among children with disabilities living with families in Zambia and receiving services at selected health centers and community-based rehabilitation centers.

**Methods.** We analyzed cross-sectional program data from 483 children aged birth to 10 years with reported special healthcare needs, collected between June 2017 and August 2021 at three health centers and ten community-based rehabilitation centers. Data included demographics, weight, height, mid-upper arm circumference (MUAC), hemoglobin concentrations, risk for feeding difficulties, and reported caregiver feeding practices. Undernutrition was assessed using the World Health Organization $z$-scores for anthropometric measurements, MUAC cut-off points, and anemia criteria. ANOVA and Pearson's chi-squared tests were used to compare groups.

**Results.** More than half of the children were boys (54.2%) and under 5 years old (59.2%). Cerebral palsy was the most common special healthcare need (63.6%), followed by an unspecified "other" category (18.6%). Undernutrition was prevalent, with 62.8% underweight, 68.1% stunted, 22.6% wasted (using MUAC), and 60.5% anemic. Children with cerebral palsy had consistently lower anthropometric $z$-scores and higher rates of stunting, wasting, and underweight compared to children with other developmental disabilities or special healthcare needs. A risk for feeding difficulties was identified in 89.5% of the children, with common issues including coughing or choking and prolonged mealtimes. Despite these challenges, most caregivers reported practicing responsive feeding techniques at least some of the time.

**Conclusion.** These findings highlight the significant risk of undernutrition and feeding difficulties among children with disabilities, underscoring the urgent need for disability-inclusive, community-based nutrition strategies in Zambia. Stakeholders should use

this information to strengthen nutrition programs and policies that uphold the rights of children with disabilities.

# INTRODUCTION

Globally, over 240 million children have disabilities (*UNICEF, 2021*). The United Nations Convention on the Rights of Persons with Disabilities (UNCRPD) defines persons with disabilities as: "…those who have long-term physical, mental, intellectual or sensory impairments, which in interaction with various barriers may hinder their full and effective participation in society on an equal basis with others" (*The United Nations, 2006*). Developmental disabilities often appear in infancy or childhood, presenting as delays in developmental milestones or impairments in one or more domains, such as cognition, motor skills, vision, hearing, speech, and behavior (*World Health Organization, 2012*).

The link between undernutrition and disability is significant: undernutrition can cause or worsen impairments, while certain disabilities can place children at higher risk of undernutrition due to factors like increased nutrient requirements, nutrient loss, reduced food intake (*Groce et al., 2014*; *Kerac et al., 2014*), and feeding difficulties (*Klein et al., 2023*). Across the world, children with disabilities also face significant and multiple barriers—attitudinal, physical, communication, and financial—that prevent them from accessing a level of healthcare and services equal to that of children without disabilities (*UNICEF, 2022*). In low- and middle-income countries, they are 34% more likely to be stunted, 25% more likely to be wasted, 25% less likely to receive early stimulation (*Hume-Nixon & Kuper, 2018*; *UNICEF, 2021*), and twice as likely to die from malnutrition during childhood (*Kuper & Heydt, 2019*). Children with disabilities are more likely to be separated from their families and placed into residential care institutions. Institutionalization, which is associated with poor growth and developmental outcomes (*van IJzendoorn et al., 2020*), is frequently driven by malnutrition and the lack of adequate community-based support for these children (*Goldman et al., 2020*). This is in disagreement with the United Nations Convention on the Rights of the Child (UNCRC), which affirms children's rights to heath, family, and non-discrimination (*The United Nations, 1989*).

In Zambia, people with disabilities experience daily challenges that infringe on their human rights, such as inaccessible environments, stigma and discrimination, and limited support (*Scherer et al., 2024*). A 2015 national survey found that 4.4% of children 2–17 years old in Zambia have functional difficulties—a reliable indicator of disability—across one or more developmental domains, according to the Washington Group Child Functioning Module (*Ministry of Community Development and Social Services, 2015*). Zambia's national policies emphasize both child nutrition and disability rights. The country has ratified key international agreements, including the UNCRC (*The United Nations, 1989*) and

UNCRPD (*The United Nations, 2006*), and passed the Persons with Disability Act No. 6 of 2012 (*Parliament of Zambia, 2012*). These commitments affirm the right to health for *all* children, including children with disabilities.

Zambia has also made malnutrition a priority in its 8th National Development Plan, aiming to reduce underweight prevalence from 11.8% in 2018 to 8.0% by 2026, wasting from 4.2% to 3.0%, and stunting from 34.6% to 25.0% (*Ministry of Finance and National Planning, 2022*). Although Zambia has made progress towards the Sustainable Development Goal (SDG) to end hunger, achieve food security and improved nutrition and promote sustainable agriculture, many SDG targets remain unmet (*Sachs, Lafortune & Fuller, 2024*). Additionally, the National Disability Policy envisions that by 2030, persons with disabilities will enjoy equal opportunities that are fundamental for living and development. Zambia has embraced the SDG principle of "Leave No One Behind" in this effort (*The United Nations Sustainable Development Group, 2015*).

Considerable data gaps exist globally, including in Zambia, regarding the nutritional status of children with disabilities. For instance, the 2018 Zambia Demographic Health Survey (DHS) (*Zambia Statistics Agency, Ministry of Health (MOH) Zambia & ICF, 2019*) reported high levels of undernutrition among children under five, but did not disaggregate data by disability status or functional difficulties. Similarly, other national nutrition surveys (*UNICEF, 1999*) have failed to collect data specific to children with disabilities, perpetuating these gaps. However, one study focusing on children in residential care institutions in Zambia showed high rates of undernutrition, especially among those with special healthcare needs such as cerebral palsy (*Makhoul et al., 2024*).

This study builds on existing data by addressing gaps in the literature on to the nutritional needs of children with disabilities in low- and middle-income countries, particularly Zambia. While previous research has focused on undernutrition, there has been limited exploration of the intersection between disability, feeding difficulties, and undernutrition in this context. Additionally, the study supports Zambia's inclusion efforts and identifies areas where current interventions may need improvement or expansion.

This study aimed to estimate the prevalence of undernutrition and feeding difficulties, and describe feeding practices among children with reported special healthcare needs. The findings from this study could have implications for shaping and strengthening policies and programs for children with disabilities in Zambia.

# MATERIALS AND METHODS

## Study design and participants

We performed a secondary analysis of de-identified cross-sectional data collected between June 2017 and August 2021 through the Improving Feeding and Nutrition Program, implemented in Zambia by SPOON and Access to Health Zambia (formerly Catholic Medical Mission Board Zambia). Launched in 2017, the program aimed to improve feeding and nutrition outcomes for two groups particularly vulnerable to undernutrition: children with disabilities and children in residential care institutions, both of whom are frequently overlooked by mainstream nutrition programs and policies.

In this study, we chose to use the term "children with special healthcare needs" to encompass not only physical, intellectual, and developmental disabilities but also health conditions or impairments that could potentially lead to disability. This approach aligns with the World Health Organization (WHO) International Classification of Functioning, Disability, and Health framework (*World Health Organization, 2001*), enabling us to capture a broader population, including those whose health conditions, when combined with environmental factors, may restrict their ability to fully participate in daily activities and social interactions. Additionally, this study focused on children with reported special healthcare needs living in family care, while data on children with and without special healthcare needs living in residential care institutions were published separately (*Makhoul et al., 2024*).

The study included a convenience sample of 483 children, ranging from birth to 10 years old, who were receiving services at one of 13 participating facilities, regardless of their specific condition. These included ten community-based rehabilitation centers (nine in Lusaka and one in Kafue) and three health centers (two in Lusaka and one in Kafue).

## Sample size estimation

Our target sample size was 400, calculated to estimate an underweight prevalence of 62.0% with a 5% margin of error and a 95% confidence interval, based on Count Me In data for children with reported special healthcare needs in Zambia. Statistical significance was set at a $p$-value of $< 0.05$.

## Study measurements and procedures of data collection

Child demographics (sex, date of birth, and reported special healthcare needs), anthropometric measurements, blood hemoglobin concentrations, and feeding data were collected by trained staff working at the participating health and community-based centers using Count Me In, a digital health app developed by *SPOON (2023)*.

### Reported special healthcare needs

Special healthcare needs were recorded in Count Me In by facility staff based on report by a primary caregiver or the staff's own knowledge of the child's diagnosis, which is typically documented in the child's records. These conditions were not clinically diagnosed by healthcare professionals as part of this study. As previously described (*Makhoul et al., 2024*), the app provided a predefined list of conditions associated with an increased risk for nutrition and feeding difficulties, from which staff could select. The conditions listed included autism spectrum disorder, cerebral palsy, cleft lip/palate, cognitive impairment, Down syndrome, HIV/AIDS, heart disease/defect, hydrocephalus, seizure disorder/epilepsy, visual impairment, low birth weight (<2.5 kg), premature birth, and a category for other conditions.

### Anthropometric measurements

Anthropometric measurements were obtained for 385 children (79.7%). Children with anthropometric data were significantly older on average (53.3 ± 1.6 months) than those without (38.2 ± 5.6 months; $p < 0.05$) and were more frequently seen in community-based centers compared to health centers (85.4% *vs.* 53.1%; $p < 0.001$). The disparity in

missing data in health centers *vs.* community-based centers may be due to several factors: community centers have a manageable workload, making staff more likely to complete measurements, while health centers, which serve children with more severe disabilities and operate high-volume clinics, may lack sufficient time or staff for anthropometric assessments. Additionally, children at community centers tend to be older, as these centers serve a broader age range. There were no differences in sex or the presence of cerebral palsy between these groups.

Anthropometric data were collected as previously described (*Makhoul et al., 2024*). Specifically, weight, recumbent length for children <24 months old, height for children ≥ 24 months old, and mid-upper arm circumference (MUAC) were measured using calibrated growth equipment and standardized techniques (*Cashin & Oot, 2018*), with data entered into Count Me In. Before measuring length or height, the study team first answered two Yes/No questions in the app: "Can the child stand without assistance?" and "Can the child fully straighten their legs?" If either question was answered "No", the app concealed the length/height entry field and displayed only the weight and MUAC fields. As a result, length and height measurements were not taken if they could not be safely or accurately completed due to conditions like contractures ($n = 20$). The app then generated $z$-scores for length/height-for-age (L/HAZ), weight-for-length/height (WL/HZ), weight-for-age (WAZ), MUAC-for-age (MUACZ) or MUAC cut-off points, based on the WHO growth standards (*World Health Organization, 2006*).

Undernutrition indicators—stunting, wasting, and underweight—were assessed using L/HAZ, WL/HZ or MUACZ for children <5 years, and WAZ, respectively. The severity of undernutrition was classified based on $z$-score thresholds as normal ($\geq -2$), moderate ($< -2$ and $\geq -3$), and severe ($< -3$) (*World Health Organization, 2006*). In addition, age-adjusted MUAC cut-offs were used to report wasting levels as follows: for ages 6-59 months: normal ($\geq 12.5$ cm), moderate ($< 12.5$ cm and $\geq 11.5$ cm), and severe ($< 11.5$ cm); and for ages 60-120 months: normal ($\geq 14.5$ cm), moderate ($< 14.5$ and $\geq 13.5$ cm), and severe ($< 13.5$ cm) (*World Health Organization & UNICEF, 2009*).

### Anemia screening

Hemoglobin concentrations were measured in a subsample of 86 children (17.8%) at three participating health and community-based centers. There were no significant differences in age, prevalence of cerebral palsy, feeding difficulties, or sex between this subsample and children who did not have a hemoglobin measurement. However, a higher percentage was screened at health centers compared to community-based centers (66.3% *vs.* 11.3%, $p < 0.001$).

The procedure for anemia screening was described elsewhere (*Makhoul et al., 2024*). Briefly, anemia screening involved measuring hemoglobin in capillary blood using the Hemocue Hb201+ analyzer (*HemoCue, 2023*). Blood was collected *via* a heel prick for children <12 months old or a finger prick for children ≥12 months using a sterile safety lancet with a retractable needle. After wiping the first 1–2 blood drops, a 10 μL sample was collected in a Hemocue Hb201+ microcuvette. The filled microcuvette was then inserted into the Hemocue analyzer, which provided hemoglobin values in g/dL. Anemia
was classified using age specific cut-offs: for 6-23 months: mild (9.5–10.4 g/dL), moderate (7.0–9.49 g/dL), and severe (<7.0 g/dL); for 24-59 months: mild (10.0–10.9 g/dL), moderate (7.0–9.9 g/dL), and severe (<7.0 g/dL); and for 5–11 years: mild (11.0–11.4 g/dL), moderate (8.0–10.9 g/dL), and severe (<8.0 g/dL) (*World Health Organization, 2024*). Facility staff captured the presence of infection (*e.g.*, malaria, fever) because hemoglobin concentrations are altered by the acute phase response to infection (*Bresnahan et al., 2014*). No infections were reported in the study sample.

### Screening for feeding difficulties

Feeding difficulties are a wide range of delays or issues that lead to oral intake that is not age-appropriate. Feeding difficulties were screened in 344 children (71.2%). These children were more likely to have cerebral palsy compared to those not screened (66.6% *vs.* 56.1%, *p* < 0.05) and were more often seen in health centers compared to community-based centers (24.4% *vs.* 13.0%, *p* < 0.01). There were no significant differences in age or sex distribution between those screened and not screened for feeding difficulties.

Screening for feeding difficulties was conducted as previously described (*Makhoul et al., 2024*). Specifically, facility staff screened children for risk of feeding difficulties by asking primary caregivers about five mealtime domains (*Bell et al., 2019*; *Dodrill & Gosa, 2015*; *Jadcherla, 2016*; *Arvedson, Brodsky & Lefton-Greif, 2020*; *Stallings et al., 1993*; *Thoyre et al., 2018*): *Assistance* (support needed to eat or drink: full, some, or none), *Tools* (utensils used: breast for children <24 months old, bottle, spoon, cup, and/or fingers), *Texture* (food consistency: formula milk, breast milk for children <24 months old, thin liquid other than formula, puree, mashed, soft and bite-sized, and/or regular foods), *Duration* (mealtime length: <10 min, 10–30 min, or >30 min), and *Frequent coughing/choking* (yes or no).

Count Me In compared the responses to developmental expectations for the child's age at the time of screening. Deviations from the following parameters indicated a risk for feeding difficulties: for ages 0–6 months—full assistance, feeding with bottles and/or breast only, and exclusive consumption of infant formula milk and/or breast milk; for ages 6–12 months—full or some assistance, feeding with bottles, breast, spoons, cups, and/or fingers, and consumption of infant formula and/or breast milk, other thin liquids, purees, mashed, and soft and bite-sized foods; for ages ≥ 12 months—some or no assistance, feeding with breast, spoons, cups, or fingers, and consumption of regular foods; and for all ages—completing mealtime in 10–30 min and not frequently coughing or choking.

Children with cerebral palsy or Down syndrome were automatically considered at risk for feeding difficulties given the increased association of these conditions with impaired feeding skills (*Rabaey, 2017*). For these children, facility staff collected additional information on tools, food textures, and frequency of coughing/choking.

### Assessment of feeding best practices

Facility staff assessed feeding best practices for children ≥ 12 months old using a questionnaire in Count Me In. The questions aimed to determine how frequently (*always*, *sometimes*, or *never*) caregivers adhered to best practices that support safety, efficiency, skill-building and social development for children with special healthcare needs (*Arts-Rodas & Benoit, 1998*; *Benjasuwantep, Chaithirayanon & Eiamudomkan, 2013*; *Vazir et al., 2013*;
*Wuehler, Hess & Brown, 2011*). The feeding practices of interest included responding to children's cues to pace feeding appropriately, interacting with children during meals, sitting at eye level while feeding, having children sit together at mealtime, and using small, age-appropriate spoons that fit comfortably in their mouths.

## Data management

Deidentified data from Count Me In were imported into STATA (version 17, StataCorp, College Station, TX) for statistical analysis. The data were checked for completeness and outliers using exploratory data analysis methods (*Tukey, 1977*). Extreme anthropometric $z$-scores and related growth measurements were excluded according to WHO recommendations for identifying biologically implausible values (*World Health Organization & UNICEF, 2019*), resulting in the removal of 43 observations from 33 children: L/HAZ: $<-6$ or $>+6$ ($n = 27$), WL/HZ: $<-5$ or $>+5$ ($n = 3$), and WAZ: $<-6$ or $>+5$ ($n = 16$). The demographics for all children with outlier $z$-scores are provided in the supplementary materials. Children with WAZ outliers were significantly older on average ($75.9 \pm 6.9$ months) than those without outliers ($52.1 \pm 1.6$ months; $p < 0.01$). There were no differences between the two groups in age for L/HAZ or WL/HZ or in sex, diagnosis, or place of assessment for WAZ, L/HAZ or WL/HZ.

Special healthcare needs (*e.g.*, cerebral palsy) were coded as 0 = absent or 1 = present. The total number of special healthcare needs was calculated by summing the presence of each condition. Children were then grouped into four categories based on their reported conditions: (1) cerebral palsy, (2) Down syndrome, (3) other developmental disabilities (autism spectrum disorder, cognitive impairment, heart disease/defect, hydrocephalus, seizure disorder/epilepsy, visual impairment), or (4) other special healthcare needs (cleft lip/palate, HIV/AIDS, low birth weight ($<2.5$ kg), premature birth, other). Children with cerebral palsy and Down syndrome were categorized separately because these two developmental disabilities have the potential to directly alter growth in children (*Cronk, 1978*; *Stallings et al., 1993*).

## Statistical analysis

We conducted exploratory analysis to assess differences between groups, with statistical significance set at a $p$-value of $<0.05$ without adjustment for multiple testing. Means $\pm$ standard deviations and frequencies were calculated for continuous and categorical variables, respectively. ANOVA was used to test for differences in continuous variables across the special healthcare needs groups. Student $t$-tests were used to compare values between two groups. Pearson's chi-squared tests were used to compare categorical variables between children with different special healthcare needs.

## Ethical considerations

Research approvals were obtained from the University of Zambia's Humanities and Social Science Research Ethics Committee, Lusaka, Zambia (HSSREC reference number: 2023-MAR-010B), the Institutional Review Boards at St. Catherine's University, St. Paul, Minnesota, USA (protocol number: 1899), and the Colorado Multiple Institutional Review Board at the University of Colorado, Aurora, Colorado, USA (submission ID: APP001-1).

The Ministry of Community Development and Social Services, responsible for overseeing the wellbeing of children with disabilities, also granted permission to collect data. As this was a retrospective analysis of existing, de-identified program data, no written informed consent was required. The study was conducted in accordance with the Declaration of Helsinki (*World Medical Association, 2022*).

## RESULTS

In this sample, boys made up a larger proportion than girls (54.2% *vs.* 45.8%). The average age of the children was $53.1 \pm 30.6$ months, with the majority (87.2%) being older than 24 months. Cerebral palsy was reported in 63.6% of the children, while 19.2% had other developmental disabilities and 26.1% had other special healthcare needs. The majority of children (89.7%) had one reported special healthcare need. Most children (78.8%) were assessed at a community-based rehabilitation center, while 21.1% were assessed at a health center (Table 1).

There was no significant difference in the mean anthropometric $z$-scores between boys and girls. Among children <120 months of age, 62.8% were underweight and 68.1% were stunted. There was a significant difference ($p < 0.05$) in mean WAZ and L/HAZ between age groups, with $z$-scores decreasing notably after 11 months of age (Table 2). For children 0–59 months of age, wasting was identified in 27.4% based on WL/HZ and 27.3% based on MUACZ, while 22.6% of children 6–120 months of age had wasting based on MUAC cut-off points. There was a significant difference by age ($p < 0.05$) in mean WL/HZ, but not in mean MUACZ, among children 0–59 months old. Additionally, wasting prevalence, based on MUAC cut-off points, significantly decreased across age groups in children 6–59 months old, but then increased in children 60 months and older ($p < 0.001$) (Table 3).

Children with cerebral palsy consistently had lower anthropometric $z$-scores compared to children with other developmental disabilities or special healthcare needs, with a higher percentage identified with stunting (74.0%), being underweight (70.7%) (Table 2), and wasting based on WL/HZ (32.3%) (Table 3). Additionally, children assessed at health centers exhibited more indicators of undernutrition than those assessed at community-based centers (Tables 2 & 3).

Among children with cerebral palsy, no significant differences in mean anthropometric $z$-scores were observed based on sex, age, or the presence of additional special healthcare needs. However, 42.9% of children with cerebral palsy aged 6–11 months were identified with wasting based on MUAC cut-offs. This prevalence was significantly higher than in other age groups ($p = 0.001$), though the finding is based on a small sample of just seven children in the 6–11 months age group (Table 4).

In the hemoglobin subsample, 60.5% were identified as having anemia (Table 5). The prevalence of anemia varied by age group ($p < 0.01$), with rates of 20%, 100%, 51.7%, 57.9% in children aged 6–11 months, 12–23 months, 24–59 months, and 60–120 months, respectively. Additionally, a higher percentage of children screened at health centers had anemia compared with those screened at community-based centers (70.2% *vs.* 41.4%, $p < 0.05$). There were no significant differences in anemia prevalence between sexes or based on reported special healthcare needs (Table 5).

**Table 1   Characteristics of children with reported special healthcare needs.**

|  | N | n (%) |
|---|---|---|
| Sex | 483 |  |
| Female |  | 221 (45.8) |
| Male |  | 262 (54.2) |
| Age at anthropometric assessment | 387 |  |
| Mean age (months) |  | 53.1 ± 30.6 |
| Age range (months) |  | 0.5-119.4 |
| Age by group |  |  |
| <6 months |  | 5 (1.3) |
| 6–11 months |  | 16 (4.1) |
| 12–23 months |  | 67 (17.3) |
| 24–59 months |  | 141 (36.4) |
| 60–120 months |  | 158 (40.8) |
| Age at anemia assessment | 86 |  |
| Mean age (months) |  | 54.9 ± 32.3 |
| Age range (months) |  | 6.4–116.7 |
| Age by group |  |  |
| <6 months |  | – |
| 6–11 months |  | 5 (5.8) |
| 12–23 months |  | 14 (16.3) |
| 24–59 months |  | 29 (33.7) |
| 60–120 months |  | 38 (44.2) |
| Reported special healthcare needs[*] | 483 |  |
| Cerebral palsy |  | 307 (63.6) |
| Down syndrome |  | 23 (4.8) |
| Other developmental disabilities |  |  |
| Cognitive impairment |  | 17 (3.5) |
| Seizure disorder/ epilepsy |  | 25 (5.2) |
| Visual impairment |  | 4 (0.8) |
| Autism spectrum disorder |  | 16 (3.3) |
| Hydrocephalus |  | 28 (5.8) |
| Heart disease/defect |  | 3 (0.6) |
| Other special healthcare needs |  |  |
| HIV/AIDS |  | 3 (0.6) |
| Cleft lip/palate |  | 2 (0.4) |
| Low birth weight |  | 25 (5.2) |
| Premature birth |  | 6 (1.2) |
| Other |  | 90 (18.6) |
| Number of reported conditions per child | 483 |  |
| One |  | 433 (89.7) |
| Two |  | 37 (7.7) |
| Three or more |  | 13 (2.7) |

**Table 1** (*continued*)

|  | N | n (%) |
|---|---|---|
| Place of assessment | 483 |  |
| Community center |  | 381 (78.9) |
| Health center |  | 102 (21.1) |

**Notes.**
*Special healthcare needs were reported by either the primary caregiver or facility staff, with no medical doctor confirming the diagnoses during the study. Conditions under "Other" were not specified.

**Table 2  Prevalence of underweight and stunting among children with reported special healthcare needs.**

|  | Underweight (0–120 months) | | | | Stunting (0–120 months) | | | |
|---|---|---|---|---|---|---|---|---|
|  | WAZ N = 371 | | | | L/HAZ N = 310 | | | |
|  | N (%) | $p$ | mean ± SD | $p$ | N (%) | $p$ | mean ± SD | $p$ |
| Overall |  | – |  | <0.001 |  | – |  | <0.001 |
| No undernutrition | 138 (37.2) |  | −0.81 ± 0.94 |  | 99 (31.9) |  | −0.86 ± 0.98 |  |
| Moderate | 74 (20.0) |  | −2.45 ± 0.28 |  | 71 (22.9) |  | −2.50 ± 0.30 |  |
| Severe | 159 (42.9) |  | −4.11 ± 0.76 |  | 140 (45.2) |  | −4.20 ± 0.77 |  |
| Sex |  | 0.236 |  | 0.609 |  | 0.679 |  | 0.240 |
| Female | 111/168 (66.1) |  | −2.60 ± 1.66 |  | 97/145 (66.9) |  | −2.63 ± 1.61 |  |
| Male | 122/203 (60.1) |  | −2.51 ± 1.67 |  | 114/165 (69.1) |  | −2.85 ± 1.67 |  |
| Age |  | 0.402 |  | 0.033 |  | 0.072 |  | 0.011 |
| <6 months | 2/4 (50.0) |  | −1.72 ± 1.01 |  | 3/4 (75.0) |  | −3.08 ± 2.51 |  |
| 6–11 months | 7/16 (43.8) |  | −1.55 ± 1.68 |  | 8/16 (50.0) |  | −1.74 ± 1.90 |  |
| 12–23 months | 41/67 (61.2) |  | −2.29 ± 1.56 |  | 40/61 (65.6) |  | −2.82 ± 1.42 |  |
| 24–59 months | 94/140 (67.1) |  | −2.73 ± 1.54 |  | 88/114 (77.2) |  | −3.07 ± 1.52 |  |
| 60–120 months | 89/144 (61.8) |  | −2.63 ± 1.79 |  | 72/115 (62.6) |  | −2.51 ± 1.73 |  |
| Reported special healthcare needs |  | <0.001 |  | <0.001 |  | 0.002 |  | 0.001 |
| Cerebral palsy | 169/239 (70.7) |  | −2.87 ± 1.66 |  | 142/192 (74.0) |  | −2.97 ± 1.67 |  |
| Down syndrome | 8/18 (44.4) |  | −1.71 ± 1.65 |  | 13/18 (72.2) |  | −2.68 ± 1.51 |  |
| Other developmental disabilities | 18/53 (34.0) |  | −1.51 ± 1.45 |  | 21/47 (44.7) |  | −1.93 ± 1.56 |  |
| Other special healthcare needs | 38/61 (62.3) |  | −2.44 ± 1.38 |  | 35/53 (66.0) |  | −2.67 ± 1.45 |  |
| Place of assessment |  | 0.002 |  | 0.003 |  | 0.063 |  | 0.015 |
| Community center | 189/317 (59.6) |  | −2.44 ± 1.69 |  | 175/265 (66.0) |  | −2.63 ± 1.68 |  |
| Health center | 44/54 (81.5) |  | −3.16 ± 1.33 |  | 36/45 (80.0) |  | −3.29 ± 1.25 |  |

**Notes.**
Classification of underweight and stunting was based on $z$-score thresholds: normal ($\geq-2$), moderate ($<-2$ and $\geq-3$), and severe ($<-3$). WAZ: Weight-for-age $z$-score; L/-HAZ: Length/Height-for-age $z$-score. Statistical analyses for categorial variables = Pearson's chi-squared and for continuous variables = Student $t$-tests and one-way ANOVA. P-values shown in bold are statistically significant.

Overall, 89.5% of the children were identified as being at risk for feeding difficulties. All children with cerebral palsy and Down syndrome were automatically classified as at risk. A high proportion of children with other developmental disabilities (71.1%) and those with other special healthcare needs (56.9%) were similarly identified. Despite these risks, 96.8% of the children were fed using age-appropriate utensils, 87.8% received food with age-appropriate textures, and 75.3% were allowed appropriate mealtime durations. Notably, one in three children experienced coughing or choking during feeding.

**Table 3 Prevalence of wasting among children with reported special healthcare needs.**

| | Wasting (0–59 months) | | | | | | | Wasting (6–120 months) | | | |
| | WL/HZ N = 201 | | | | MUACZ N = 205 | | | | MUAC (cm) N = 340 | | |
| | N (%) | p | mean ± SD | p | N (%) | p | mean ± SD | p | N (%) | p | mean ± SD | p |
|---|---|---|---|---|---|---|---|---|---|---|---|---|
| Overall | | – | | <0.001 | | – | | <0.001 | | – | | <0.001 |
| No undernutrition | 146 (72.6) | | −0.40 ± 1.19 | | 149 (72.7) | | −0.55 ± 1.07 | | 263 (77.4) | | 15.2 ± 1.7 | |
| Moderate | 29 (14.4) | | −2.35 ± 0.23 | | 36 (17.6) | | −2.58 ± 0.22 | | 38 (11.2) | | 12.9 ± 1.0 | |
| Severe | 26 (12.9) | | −3.61 ± 0.50 | | 20 (9.8) | | −3.72 ± 0.55 | | 39 (11.5) | | 11.5 ± 1.0 | |
| Sex | | 0.644 | | 0.150 | | 0.637 | | 0.199 | | 0.979 | | 0.084 |
| Female | 28/97 (28.9) | | −1.26 ± 1.46 | | 28/97 (28.9) | | −1.35 ± 1.47 | | 35/155 (22.6) | | 14.3 ± 2.0 | |
| Male | 27/104 (26.0) | | −0.94 ± 1.67 | | 28/108 (25.9) | | −1.09 ± 1.45 | | 42/185 (22.7) | | 14.7 ± 2.1 | |
| Age | | 0.304 | | 0.016 | | 0.358 | | 0.653 | | <0.001 | | <0.001 |
| <6 months | 0/4 (0.0) | | 1.11 ± 1.90 | | 0/3 (0.0) | | −0.44 ± 0.51 | | – | | – | |
| 6–11 months | 2/16 (12.5) | | −0.68 ± 1.42 | | 6/14 (42.9) | | −1.35 ± 1.35 | | 7/14 (50.0) | | 12.8 ± 1.3 | |
| 12–23 months | 19/64 (29.7) | | −1.04 ± 1.58 | | 19/65 (29.2) | | −1.34 ± 1.57 | | 19/65 (29.2) | | 13.3 ± 1.8 | |
| 24–59 months | 34/117 (29.1) | | −1.25 ± 1.53 | | 31/123 (25.2) | | −1.15 ± 1.44 | | 13/126 (10.3) | | 14.4 ± 1.9 | |
| 60–120 months | – | | – | | – | | – | | 38/135 (28.1) | | 15.4 ± 2.0 | |
| Reported special healthcare needs | | 0.059 | | 0.002 | | 0.005 | | 0.001 | | 0.003 | | <0.001 |
| Cerebral palsy | 41/127 (32.3) | | −1.29 ± 1.65 | | 34/128 (26.6) | | −1.25 ± 1.40 | | 48/214 (22.4) | | 14.5 ± 1.9 | |
| Down syndrome | 2/11 (18.2) | | −0.68 ± 1.19 | | 0/11 (0.0) | | −0.24 ± 1.03 | | 1/18 (5.6) | | 16.2 ± 2.2 | |
| Other developmental disabilities | 1/21 (4.8) | | 0.07 ± 1.24 | | 3/23 (13.0) | | −0.50 ± 1.13 | | 6/50 (12.0) | | 15.3 ± 1.8 | |
| Other special healthcare needs | 11/42 (26.2) | | −1.20 ± 1.31 | | 19/43 (44.2) | | −1.76 ± 1.65 | | 22/58 (37.9) | | 13.7 ± 2.2 | |
| Place of assessment | | 0.016 | | 0.034 | | <0.001 | | <0.001 | | <0.001 | | <0.001 |
| Community center | 39/164 (23.8) | | −0.98 ± 1.57 | | 32/167 (19.2) | | −0.95 ± 1.40 | | 49/287 (17.1) | | 14.9 ± 2.0 | |
| Health center | 16/37 (43.2) | | −1.59 ± 1.53 | | 24/38 (63.2) | | −2.37 ± 1.14 | | 28/53 (52.8) | | 12.8 ± 1.5 | |

**Notes.**
Classification of wasting was based on z-score thresholds: normal ($\geq-2$), moderate ($<-2$ and $\geq-3$), and severe ($<-3$) or MUAC cut-off points: for ages 6–59 months: normal ($\geq12.5$ cm), moderate ($<12.5$ cm and $\geq11.5$ cm), and severe ($<11.5$ cm); and for ages 60–120 months: normal ($\geq14.5$ cm), moderate ($<14.5$ and $\geq13.5$ cm), and severe ($<13.5$ cm). MUAC: Mid-upper arm circumference; WL/HZ: Weight-for-length/height z-score; MUACZ: MUAC-for-age z-score. Statistical analyses for categorial variables = Pearson's chi-squared and for continuous variables = Student t-tests and one-way ANOVA.
P-values shown in bold are statistically significant.

Five feeding practices for children $\geq$ 12 months old were assessed, with most recommended practices reported by caregivers as being consistently followed (Fig. 1). Most caregivers (87.4%) reported always following their children's feeding cues, while fewer caregivers indicated that they followed the cues sometimes (10.7%) or never (1.9%). Small, age-appropriate spoons were used consistently for 81.6% of the children, sometimes for 14.9%, and never for 3.5%. Interacting with the children through singing or talking was a common practice, with 76.6% of caregivers doing so always, 12.2% sometimes, and 11.2% never. Two-thirds of caregivers (65.7%) always sat at eye level with their child during mealtime, while 2.9% did so sometimes and 31.4% never did. Having the child sit with others for mealtime was the least common practice, though 57.8% of children always sat with others, 18% sometimes, and 24.2% never.

No significant differences were observed based on the sex of the child in caregivers' adherence to the five recommended feeding practices. However, significant differences were seen depending on the presence of cerebral palsy, age, and place of assessment.
**Table 4  Prevalence of undernutrition among children with reported cerebral palsy.**

| | Underweight (0–120 months) WAZ n = 239 | | Stunting (0–120 months) L/HAZ n=192 | | Wasting (0–59 months) | | | | Wasting (6–120 months) MUAC (cm) n = 214 | |
| | | | | | WL/HZ n=127 | | MUACZ n=128 | | | |
| | N (%) | p | N (%) | p | N (%) | p | N (%) | p | N (%) | p |
|---|---|---|---|---|---|---|---|---|---|---|
| Overall | | – | | – | | – | | – | | – |
| No undernutrition | 70 (29.3) | | 50 (26.0) | | 86 (67.7) | | 94 (73.4) | | 166 (77.6) | |
| Moderate | 47 (19.7) | | 40 (20.8) | | 19 (15.0) | | 24 (18.8) | | 24 (11.2) | |
| Severe | 122 (51.1) | | 102 (53.1) | | 22 (17.3) | | 10 (7.8) | | 24 (11.2) | |
| Sex | | 0.923 | | 0.492 | | 0.463 | | 0.805 | | 0.177 |
| Female | 76/107 (71.0) | | 63/88 (71.6) | | 20/56 (35.7) | | 14/55 (25.5) | | 17/94 (18.1) | |
| Male | 93/132 (70.5) | | 79/104 (76.0) | | 21/71 (29.6) | | 20/73 (27.4) | | 31/120 (25.8) | |
| Age | | 0.247 | | 0.167 | | 0.268 | | 0.248 | | **0.001** |
| <6 months | 2/3 (66.7) | | 3/3 (100.0) | | 0/3 (0.0) | | 0/2 (0.0) | | – | |
| 6–11 months | 3/8 (37.5) | | 5/8 (62.5) | | 1/8 (12.5) | | 3/7 (42.9) | | 3/7 (42.9) | |
| 12–23 months | 23/33 (69.7) | | 22/30 (73.3) | | 9/32 (28.1) | | 5/32 (15.6) | | 5/32 (15.6) | |
| 24–59 months | 76/101 (75.3) | | 66/81 (81.5) | | 31/84 (36.9) | | 26/87 (29.9) | | 10/89 (11.2) | |
| 60–120 months | 65/94 (69.2) | | 46/70 (65.7) | | – | | – | | 30/86 (34.9) | |
| # reported conditions in addition to CP | | 0.881 | | 0.409 | | 0.212 | | 0.276 | | 0.569 |
| CP only | 151/214 (70.6) | | 128/175 (73.1) | | 36/117 (30.8) | | 33/119 (27.7) | | 44/200 (22.0) | |
| CP + other conditions | 18/25 (72.0) | | 14/17 (82.4) | | 5/10 (50.0) | | 1/9 (11.1) | | 4/14 (28.6) | |

**Notes.**

Classification of underweight, stunting, and wasting was based on $z$-score thresholds: normal ($\geq-2$), moderate ($<-2$ and $\geq-3$), and (severe $<-3$). Wasting was also based on MUAC cut-off points: for ages 6-59 months: normal ($\geq12.5$ cm), moderate ($<12.5$ cm and $\geq11.5$ cm), and severe ($<11.5$ cm); and for ages 60-120 months: normal ($\geq14.5$ cm), moderate ($<14.5$ and $\geq13.5$ cm), and severe ($<13.5$ cm). MUAC: Mid-upper arm circumference; WAZ: Weight-for-age $z$-score; L/HAZ: Length/Height-for-age $z$-score; WL/HZ: Weight-for-length/height $z$-score; MUACZ: MUAC-for-age $z$-score. Statistical analyses for categorial variables = Pearson's chi-squared.
P-values shown in bold are statistically significant.

For example, children with cerebral palsy were less likely to sit with others during meals compared to children without cerebral palsy (48% *vs.* 71.7%, $p < 0.01$). Additionally, caregivers of children with cerebral palsy were more likely to consistently sit at eye level (72.7% *vs.* 52.8%, $p < 0.05$) and interact with the child during feeding (83.6% *vs.* 65%, $p < 0.05$) compared to caregivers of children without cerebral palsy.

The average age of children whose caregivers always interacted with them during feeding was $60.4 \pm 3.8$ months, significantly older ($p < 0.001$) than those whose caregivers did not consistently interact ($33.0 \pm 4.5$ months). Similarly, children whose caregivers fed them at eye level had a higher mean age of $62.0 \pm 4.3$ months, compared to $35.1 \pm 3.9$ months for those not always fed at eye level ($p < 0.001$).

There was no significant age difference for children who consistently used small spoons, nor for the place of assessment. However, small spoons were significantly more frequently used ($p < 0.001$) for children seen at a community-based center (68.0%) compared to those seen at a health center (25.8%). Additionally, children assessed at a community-based center

**Table 5   Prevalence of anemia among children 6–120 months with reported special healthcare needs.**

|  | Anemia | | Hemoglobin (g/dL) | |
| --- | --- | --- | --- | --- |
|  | N (%) | *p* | mean ± SD | *p* |
| Overall |  | – |  | **<0.001** |
| No anemia | 34/86 (39.5) |  | 12.4 ± 1.1 |  |
| Mild anemia | 16/86 (18.6) |  | 11.1 ± 0.6 |  |
| Moderate anemia | 21/86 (24.4) |  | 9.3 ± 0.9 |  |
| Severe anemia | 15/86 (17.4) |  | 9.1 ± 2.0 |  |
| Sex |  | 0.797 |  | 0.362 |
| Female | 29/47 (61.7) |  | 11.0 ± 2.1 |  |
| Male | 23/39 (59.0) |  | 10.7 ± 1.6 |  |
| Age at assessment |  |  |  |  |
| 6–11 months | 1/5 (20.0) | **0.003** | 10.9 ± 1.0 | **0.003** |
| 12–23 months | 14/14 (100.0) |  | 9.5 ± 1.7 |  |
| 24–59 months | 15/29 (51.7) |  | 10.6 ± 2.0 |  |
| 60–120 months | 22/38 (57.9) |  | 11.5 ± 1.6 |  |
| Reported special healthcare needs |  | 0.655 |  | 0.664 |
| Cerebral palsy | 35/58 (60.3) |  | 11.0 ± 2.0 |  |
| Down syndrome | 1/1 (100.0) |  | 11.0 |  |
| Other developmental disabilities | 1/3 (33.3) |  | 11.4 ± 1.4 |  |
| Other special healthcare needs | 15/24 (62.5) |  | 10.5 ± 1.7 |  |
| Place of assessment |  | **0.01** |  | **0.008** |
| Community center | 12/29 (41.4) |  | 11.6 ± 1.7 |  |
| Health center | 40/57 (70.2) |  | 10.5 ± 1.9 |  |

**Notes.**

Statistical analyses for categorial variables = Pearson's chi-squared and for continuous variables = Student's *t*-tests and one-way ANOVA.

P-values shown in bold are statistically significant.

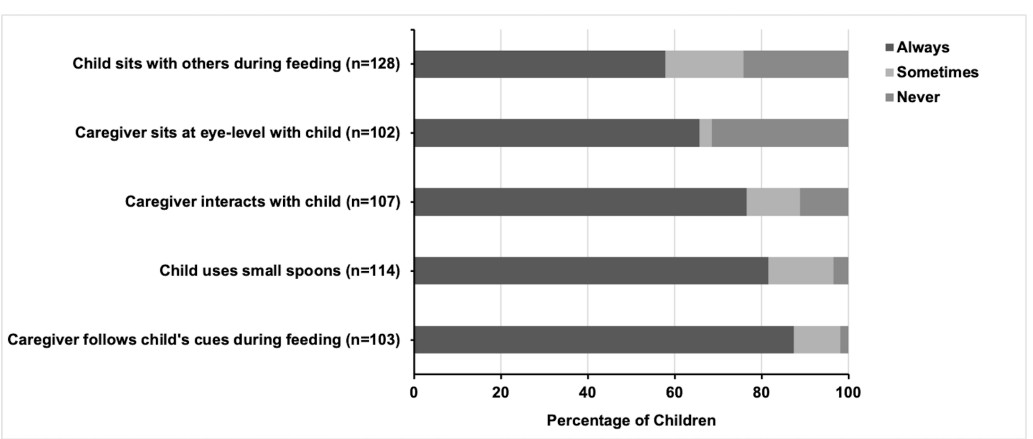

**Figure 1   Feeding best practices among children older than 12 months of age with reported special healthcare needs.**

were much more likely to eat with others (68.0%) than those seen at a health center (25.8%) ($p < 0.001$).

## DISCUSSION

This study adds to the growing body of literature on the high prevalence of undernutrition among children with disabilities in low- and middle-income countries, while also helping to fill the gap in understanding feeding difficulties within this population. Our findings highlight the critical nutritional vulnerabilities faced by children with special healthcare needs in family care settings in Zambia, with notably severe risks observed in children with cerebral palsy. The overall prevalence of undernutrition, including its most severe forms, and the risk for feeding difficulties were alarmingly high, with children with cerebral palsy at a disproportionately greater risk compared to children with other developmental disabilities or special healthcare needs. These results underscore the urgent need to accelerate efforts in Zambia to strengthen disability inclusion within nutrition surveillance, and in nutrition programs and policies.

### Nutritional status

The high prevalence of underweight and stunting, affecting two-thirds of children with reported special healthcare needs, represents a very high public health concern (*De Onis et al., 2019*). The co-occurrence of severe undernutrition, high rates of wasting, and widespread anemia—particularly among children under five years of age—indicates chronic nutritional deficiencies with potentially lasting developmental and health impacts (*Black et al., 2008*).

Undernutrition was disproportionately higher among children with disabilities under five years of age compared to the general population. The prevalence rates for stunting (71.3%), underweight (63.4%), wasting (27.4%), and anemia (62.5%) were markedly higher than those reported in the 2018 DHS, where stunting was 34.6%, underweight 11.8%, wasting 4.2%, and anemia 58.0% (*Zambia Statistics Agency, Ministry of Health (MOH) Zambia & ICF, 2019*). DHS data were not disaggregated by disability, leaving it unclear whether, or how many, children with disabilities were included in these national estimates (*Zambia Statistics Agency, Ministry of Health (MOH) Zambia & ICF, 2019*). As a result, the true national figures could potentially be higher. The rates observed in this study also far exceed the targets set in Zambia's 8th National Development Plan (*Ministry of Finance and National Planning, 2022*), with underweight and wasting surpassing the targets by over sevenfold, and stunting more than double the national goals.

Our findings are consistent with existing research highlighting the high risk of undernutrition among children with disabilities. A meta-analysis of 17 studies from low- and middle-income countries (*Hume-Nixon & Kuper, 2018*) found that children with disabilities were three times more likely to be underweight and twice as likely to be stunted or wasted. Similarly, an analysis of UNICEF-supported Multiple Indicator Cluster Survey data from 30 low- and middle-income countries found that children with disabilities, aged 2–4 years, were significantly more likely to experience stunting, wasting, and underweight

compared to children without disabilities (*Rotenberg et al., 2023*). While both analyses included countries from Eastern and Southern Africa, neither featured data from Zambia.

We found a significantly high prevalence of moderate-to-severe undernutrition among children with cerebral palsy, with rates notably higher compared to children with other developmental disabilities and special healthcare needs. These rates were generally worse among older children with cerebral palsy and were consistent with reports from other low- and middle-income countries (*Hamid Namaganda et al., 2023*; *Jahan et al., 2021a*; *Jahan et al., 2021b*; *Jahan et al., 2021c*; *Karim et al., 2019*; *Polack et al., 2018*). In a study spanning Bangladesh, Indonesia, Nepal, and Ghana, 72–98% of children with cerebral palsy had at least one form of undernutrition, with older age, low maternal education, and severe motor impairment identified as key predictors (*Jahan et al., 2021c*). Similarly, research in Eastern Uganda showed that 64% of children and adolescents with cerebral palsy were stunted, wasted, or underweight compared to 27% of their peers without cerebral palsy, with feeding difficulties and severe motor impairments driving these differences (*Hamid Namaganda et al., 2023*). In Zambia, children with cerebral palsy living in residential care institutions also exhibited high levels of stunting (71.4%) and underweight (69.2%) (*Makhoul et al., 2024*). Another study in Zambia identified feeding difficulties, caregiver practices, and the severity of cerebral palsy as key risk factors for severe undernutrition in children with cerebral palsy (*Simpamba, 2020*).

While some degree of undernutrition is expected in children with cerebral palsy, especially those with feeding difficulties (*Fung et al., 2002*; *Stallings et al., 1993*; *Stevenson, Roberts & Vogtle, 1995*; *Strand et al., 2016*), our findings indicate that the severity of undernutrition in our sample exceeds typically expectations. Severe wasting, or severe acute malnutrition, among children with cerebral palsy under five years of age was 17.3% based on WL/HZ and 7.8% based on MUACZ. These figures are consistent with a multi-country study that reported severe wasting in 7.7–22.6% and 10.1–22% of children with cerebral palsy using WL/HZ and MUACZ, respectively (*Jahan et al., 2021c*). Cerebral palsy has been identified as a significant risk factor for hospital readmission following treatment for severe acute malnutrition in Zambia and Zimbabwe (*Bwakura-Dangarembizi et al., 2022*). Similarly, in Malawi, children with cerebral palsy faced a significantly higher risk of death from severe acute malnutrition compared to their peers without disabilities (*Lelijveld et al., 2020*), further underscoring the extreme nutritional vulnerability for this population.

Our finding of elevated anemia levels is significant, despite the small sample size, as this is likely the first study to document anemia specifically in children with disabilities in Zambia. The anemia rates we observed were higher than national averages (*Zambia Statistics Agency, Ministry of Health (MOH) Zambia & ICF, 2019*), and were notably elevated among children aged 12–23 months, consistent with other studies (*Makhoul et al., 2024*; *Sun et al., 2021*). Research focusing on children in residential care institutions in Zambia also revealed a higher prevalence of anemia among those with special healthcare needs (63.3%) compared to those without (52.0%) (*Makhoul et al., 2024*). Childhood anemia in Zambia is partly attributed to high intake of grain-based staple foods, which contain phytic acids that inhibit iron absorption. This issue is further exacerbated by a low intake of animal-based iron sources (*Kobayashi, Negi & Nakazawa, 2022*). Additionally, households with children with

disabilities in Zambia have been shown to have significantly reduced dietary diversity—a key factor in preventing anemia (*Shibeshi et al., 2024*)—compared to households without children with disabilities (*Hearst et al., 2023*).

Notably, children assessed at health centers exhibited higher rates of undernutrition across all indicators compared to those at community-based rehabilitation centers despite being, on average, 16.4 months younger and having a lower proportion of children with cerebral palsy (50% *vs.* 67%). This discrepancy may be due to the fact that health center attendees often sought care for medical issues that could have exacerbated their risk of undernutrition, or because caregivers had specific concerns about their child's nutrition.

While we did not investigate the underlying causes of undernutrition in our sample, it is known that in children with disabilities, undernutrition often results from a mix of factors: nutritional (*e.g.*, poor dietary intake, increased energy needs, feeding difficulties, and poor feeding practices), non-nutritional (*e.g.*, functional limitations, growth hormone deficiency, and medical complications), and socioeconomic factors (*e.g.*, poverty, limited access to specialized care, and inadequate caregiver support) (*Jahan et al., 2022*; *Kuperminc & Stevenson, 2008*). Our study explored feeding difficulties and caregiver practices in light of the significant stress and hardship frequently reported by caregivers of children with disabilities, especially those with feeding difficulties. Caregivers in resource-limited settings face numerous challenges including stressful mealtimes, limited resources, lack of support or training, high healthcare costs, increased caregiving burdens, and social stigma (*Dlamini, Chang & Nguyen, 2023*; *Mwinbam et al., 2023*; *Shaba et al., 2020*), underscoring the need for comprehensive interventions and support at the community and health system levels.

## Feeding difficulties and feeding practices

Up to 80% of children with disabilities experience feeding difficulties (*Lefton-Greif & Arvedson, 2007*), which, if left unaddressed, can result in chronic undernutrition, respiratory illnesses, stressful mealtimes, low quality of life for both the child and caregiver, and premature death (*Kamal et al., 2023*; *Klein et al., 2023*; *Polack et al., 2018*). In this cohort, the majority of children (89.5%) were identified as at risk for feeding difficulties. The Count Me In feeding screener used in the study automatically flagged children with cerebral palsy and Down syndrome as high-risk due to the strong association between these conditions and feeding difficulties (*Rabaey, 2017*). Among children with other developmental disabilities and special healthcare needs, 62.5% were also at risk. This rate was higher than the 41.4% reported for children with special healthcare needs in residential care institutions and the 26.0% for those without special healthcare needs (*Makhoul et al., 2024*) using the same screener. It is important to note that the feeding data in this study were based on self-reported practices by mothers and caregivers, which may introduce recall and response biases.

Coughing or choking during or after feeding was prevalent among children with cerebral palsy (39.7%), potentially signaling swallowing difficulties or increased risk for aspiration, which are strongly linked to severe undernutrition and lower anthropometric $z$-scores (*Fung et al., 2002*; *Herrera-Anaya et al., 2016*; *Simpamba, 2020*; *Strand et al., 2016*). Prolonged mealtimes—lasting over 30 min—were also common. Lengthy feeding sessions

often reflect underlying issues, such as poor oral motor skills or swallowing difficulties, leading to fatigue, frustration, and stress for both the child and the caregiver (*Arvedson, 2013*). This can reduce the child's willingness to eat, limit dietary intake, and contribute to undernutrition.

Despite these challenges, most caregivers in the study adhered to responsive feeding practices, at least some of the time. They paced feeding in response to their child's cues, and sat at eye level and engaged with the child during meals. These responsive feeding practices were more common among caregivers of children with cerebral palsy, likely reflecting the greater dependency these children have on their caregivers for feeding. However, children with cerebral palsy were less likely to sit with others during mealtime, limiting their opportunity to socialize and participate fully.

## Limitations

The results from this study may not fully represent all children with disabilities in Zambia due to several limitations. First, as this was an analysis of programmatic data not specifically designed to assess the nutritional status of children with disabilities, we could not determine the underlying factors contributing to undernutrition in this population. We were unable to account for certain potential confounders, such as socioeconomic status and severity of disability, which may have influenced nutritional outcomes. For example, children from lower socioeconomic backgrounds may have more limited access to diverse foods, healthcare, and support services, all of which could contribute to higher rates of undernutrition. Similarly, children with more severe disabilities may face greater feeding challenges, increasing their risk of undernutrition. Second, special healthcare needs were reported by primary caregivers and facility staff. While diagnoses, typically made by medical professionals, are usually documented in children's records at health or community centers, these diagnoses were not independently verified in this study. The lack of specification in the "other" category of health conditions could have resulted in the misclassification of some children within the special healthcare needs groups. Third, inaccuracies in growth measurements and their interpretations for children with disabilities, particularly cerebral palsy, are possible. Measuring length or height in children with cerebral palsy is challenging, particularly for those with contractures (*Hardy et al., 2018*; *Sørensen et al., 2021*). Although we attempted to mitigate errors by excluding measurements for children unable to stand or straighten their legs independently, measurement inaccuracies may still have occurred. While there was no significant difference in the type of diagnosis between children with $z$-score outliers and those without, children with outliers were, on average, older. Moreover, children with cerebral palsy often experience different growth patterns than their peers without cerebral palsy, so applying WHO growth standards could overestimate their prevalence of undernutrition (*Sørensen et al., 2021*). We used on WHO growth charts due to the limited applicability of cerebral palsy-specific growth charts (*Brooks et al., 2011*) in low- and middle-income countries. The literature remains inconclusive regarding the use of weight for length/height or weight for age for assessing the nutritional status of children with cerebral palsy, and MUAC may not accurately reflect their body composition (*Aydın et al., 2023*; *Hayes et al., 2023*; *Tomoum et al., 2010*). Finally, the generalizability of

our results is limited due to several factors, including the majority of study sites being located in Lusaka, sample sizes for certain special healthcare needs and age categories, and potential bias from missing data or excluded outliers. Our sample was, on average, older (53.1 ± 30.6 months), with 77.2% of children aged 24 months and older. This age distribution may reflect delayed diagnoses, stigma and other barriers preventing families from seeking services earlier, and selective enrollment of children with poor nutritional status for anthropometric measurements and support. The under-representation of younger children with anthropometric data is an important limitation and highlights the need for future studies to include a broader age range, particularly younger children with disabilities. Additionally, the missing data, which is likely missing at random, may introduce sampling bias.

## CONCLUSION AND RECOMMENDATIONS

Our study revealed unacceptably high levels of undernutrition and feeding difficulties among children with disabilities living with families in Zambia and receiving services from selected health centers and community-based rehabilitation centers. There is an urgent need for coordinated efforts to enhance disability inclusion in nutrition and feeding programs in Zambia and similar settings. We recommend the following actions:

- Update relevant policies, guidelines, and budgeting processes to prioritize children with disabilities, ensuring they are included in data collection and program planning.
- Design and implement nutrition programs that specifically include children with disabilities, incorporating both mainstream and targeted support.
- Provide comprehensive training, resources, and referral pathways for healthcare workers to deliver inclusive and effective nutrition and feeding services tailored to children with disabilities or feeding difficulties.
- Offer resources and support to families and caregivers to help them implement safe and responsive nutrition and feeding practices for children with disabilities or feeding difficulties.
- Increase awareness among communities and stakeholders about the rights of children with disabilities and the critical need to improve nutrition and feeding practices for children with disabilities.

## ACKNOWLEDGEMENTS

We acknowledge and are grateful for the contributions of the SPOON team, especially Jon Baldivieso for developing Count Me In and Lauren Hughey for supporting project management; the Access to Health Zambia team for their role in supporting implementation of Count Me In in health and community-based centers; the Improving Feeding and Nutrition Program Master Trainers, Lwiindi Kabondo, Claire Mwila, and Clement Chibuta for supporting data collection; the project Technical Advisory Group for their expert advice and guidance; and the staff, caregivers, and children who participated in this research.

### Funding

This work was funded by a grant from the GHR Foundation. The funders had no role in study design, data collection and analysis, decision to publish, or preparation of the manuscript.

### Grant Disclosures

The following grant information was disclosed by the authors:
GHR Foundation.

### Competing Interests

Zeina Makhoul, Carolyn Moore, Kate Miller, and Douglas Taren are employed by SPOON and Mulemba Ndonji, Watson Shungu Mwandileya, and Edgar Lunda are employed by Access to Health Zambia. Douglas Taren serves on SPOON's Board of Trustees. The authors declare there are no competing interests.

### Author Contributions

- Zeina Makhoul conceived and designed the experiments, analyzed the data, prepared figures and/or tables, authored or reviewed drafts of the article, and approved the final draft.
- Julie M. Long conceived and designed the experiments, analyzed the data, prepared figures and/or tables, authored or reviewed drafts of the article, and approved the final draft.
- Mulemba Ndonji performed the experiments, authored or reviewed drafts of the article, and approved the final draft.
- Carolyn Moore conceived and designed the experiments, prepared figures and/or tables, authored or reviewed drafts of the article, and approved the final draft.
- Watson Shungu Mwandileya performed the experiments, authored or reviewed drafts of the article, and approved the final draft.
- Edgar Lunda performed the experiments, authored or reviewed drafts of the article, and approved the final draft.
- Kate Miller conceived and designed the experiments, authored or reviewed drafts of the article, and approved the final draft.
- Bradley S. Miller conceived and designed the experiments, authored or reviewed drafts of the article, and approved the final draft.
- Douglas Taren conceived and designed the experiments, analyzed the data, prepared figures and/or tables, authored or reviewed drafts of the article, and approved the final draft.

### Human Ethics

The following information was supplied relating to ethical approvals (*i.e.*, approving body and any reference numbers):

The following institutions granted approval to carry out the study: University of Zambia's Humanities and Social Science Research Ethics Committee, Lusaka, Zambia (HSSREC reference number: 2023-MAR-010B), the Institutional Review Boards at St. Catherine's University, St. Paul, Minnesota, USA (protocol number: 1899), and the Colorado Multiple Institutional Review Board at the University of Colorado, Aurora, Colorado, USA (submission ID: APP001-1)

## Data Availability

The de-identified dataset used in the study is available in the Supplemental File.

## Supplemental Information

Supplemental information for this article can be found online at http://dx.doi.org/10.7717/peerj.20023#supplemental-information.

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
