# Peer review of "Undernutrition and feeding difficulties among children with disabilities in Zambia: insights for inclusive nutrition strategies"

_PeerJ, doi:10.7717/peerj.20023_

## Round 0.1 · original submission · Minor Revisions

The corrections suggested by the reviewers are fairly minor and I hope you can address them quickly.

Reviewer 1 ·

Basic reporting

This study examines undernutrition and feeding difficulties among children with disabilities in Zambia. The structure of the article is good.

In Lines 149-153, you provided a list of "predefined conditions", but it is not clear how these conditions were diagnosed. Please explain whether these conditions were diagnosed by healthcare professionals and what criteria were used.

In Lines 156-160, you mentioned that children with anthropometric data were significantly older, but you did not explain why this happened or discuss how this age difference might affect your results.

Experimental design

no comment.

Validity of the findings

The validity of the findings is challenged by several issues. You excluded 47 observations (Lines 243-247) but did not explain clearly why these were excluded. You should specify what WHO criteria you used and compare the characteristics of excluded versus included observations. This will help readers understand if the exclusions might have biased your results.

Your statistical analysis has important limitations. You only did univariate analyses without controlling for confounding factors like age, socioeconomic status, or disability severity. You should use multivariable regression to adjust for these factors. Also, you used complete case analysis for missing data, which can cause bias if data are not missing completely at random.

Reviewer 2 ·

Basic reporting

No comment, all well written and structured.

Experimental design

No comment.

Validity of the findings

No comment.

Additional comments

Thank you for the opportunity to review this paper. It is well written and clear, and handles an important topic. The analysis is simple but effective and clearly demonstrates that there needs to be action on improving the nutritional status of children with special health care needs.

My only comment would be to add the MUAC cut-off points in Table 3 and 4 for completeness, as well as add % labels for Figure 1 as while the visual is useful, the actual numbers makes it easier to compare.

·

Basic reporting

Abstract
This abstract presents a timely study on undernutrition and feeding difficulties among children with disabilities in Zambia. The use of cross-sectional data offers clear insights, particularly regarding the high prevalence of undernutrition among children with cerebral palsy. The findings effectively underscore the need for inclusive, community-based nutrition strategies.

Introduction
The extensive use of current literature lends credibility and depth, and the discussion on data gaps clearly highlights the need for further research. However, streamlining some sections could enhance clarity. For example, condensing the background information would help focus the reader on the study’s objectives. Additionally, clarifying how the study builds on existing data could further strengthen the narrative. Overall, it’s a comprehensive and well-supported introduction that sets a solid foundation for the study.

Experimental design

Materials and methods
This methods chapter is well-organized and clear. It comprehensively outlines the study design, participant selection, and data collection methods using the Count Me In app, ensuring transparency and methodological rigor. A brief discussion on potential biases from the convenience sample could further enhance the chapter. Overall, the methods are robust and appropriate for the study's aims. I appreciate how the chapter explains the operational definition of “children with special healthcare needs” and the systematic approach to data collection, including anthropometric measurements, anemia screening, and feeding difficulty assessments.

Results
This results chapter is impressively detailed and well-organized. I appreciate the thorough statistical analysis and use of tables for clarity. One point to note is the small sample size in the 6–11 months subgroup, which may affect the robustness of some findings. Overall, the results are presented in a manner that effectively supports the study's objectives.

Validity of the findings

Discussion
This discussion chapter is comprehensive and well-structured, clearly outlining the significance of the findings in the context of existing literature. The authors clearly highlight the increased risks of undernutrition and feeding difficulties among children with disabilities, particularly those with cerebral palsy, and compare these with national benchmarks and previous studies. The discussion on the relationship between nutritional status and various socioeconomic factors is insightful and emphasizes the urgent need for targeted interventions.
Conclusion
The conclusion and recommendation chapter effectively summarizes the study’s findings and calls for urgent, coordinated actions to improve nutrition and feeding practices for children with disabilities in Zambia. It provides clear recommendations across policy, program design, healthcare training, and community awareness.

Additional comments

Dear Author,

I am honored to review your manuscript and highly recommend its acceptance for publication. Your work addresses a critical topic, particularly in low- and middle-income countries where socioeconomic challenges are significant. I appreciate your focus on these vital issues, and I believe your study will make a valuable contribution to the field.

Sincerely,
Raishan

---

## Round 0.2 · Minor Revisions

Reviewer 1 ·

Basic reporting

The authors have addressed my previous concerns regarding the explanation of missing anthropometric data disparities between health centers and community-based centers, as well as the age differences in the sample.

Experimental design

No comment

Validity of the findings

The exclusion of 47 observations based on WHO criteria for biologically implausible z-scores (lines 259–264) is justified, but further transparency would strengthen the analysis. To ensure these exclusions did not systematically bias the sample, the authors should compare excluded and included cases (e.g., age, disability type, center type) in supplementary materials. For instance, if excluded children were disproportionately younger or had specific disabilities, this could skew generalizability. A brief comparison would clarify whether exclusion patterns align with known measurement challenges (e.g., severe contractures in health centers) or reflect broader systemic biases.

I recommend that the authors conduct multivariable regression analyses to adjust for potential confounders. Additionally, stratified analyses based on key confounding variables could further help in addressing confounding effects. It would also be important to discuss the potential impact of unmeasured confounding—such as socioeconomic status or disability severity—in the limitations section.

---

## Round 0.3 · accepted · Accept

The authors have addressed all the comments from the reviewers. The manuscript is ready for publication.

Reviewer 1 ·

Basic reporting

-

Experimental design

-

Validity of the findings

The authors have addressed my previous concerns about excluding data based on implausible z-scores by adding further analyses and providing more detailed explanations. They've also acknowledged the potential confounding as a limitation, which responds to my previous recommendation of multivariable regression models to address the potential confounding. I have no additional comments at this time.